# Whole Black Soldier Fly Larvae (*Hermetia illucens*) as Dietary Replacement of Extruded Feed for Tambaqui (*Colossoma macropomum*) Juveniles

**Betselene M. Ordoñez [1], Thiago M. Santana [2], Davison P. Carneiro [2], Driely K. M. dos Santos [3], Gustavo A. P. Parra [1], Luis C. C. Moreno [1], Nelson P. Teixeira Filho [4], Fredy A. A. Aguilar [5], Fernando Y. Yamamoto [6,7,*] and Ligia U. Gonçalves [8,*]**

[1] Grupo de Investigación en Biodiversidad y Desarrollo Amazónico BYDA, Universidad de la Amazonia, Caquetá 18002, Colombia

[2] Programa de Pós-graduação em Ciência Animal e Recursos Pesqueiros, Universidade Federal do Amazonas, Manaus 69067-005, AM, Brazil

[3] Programa de Pós-graduação em Aquicultura, Universidade Nilton Lins, Manaus, Amazonas 69058-030, Brazil

[4] Laboratório de Entomologia e Acarologia Agrícola, Universidade Federal do Amazonas, Manaus 69067-005, AM, Brazil

[5] Faculdad de Ciencias Pecuarias, Departamento de Zootecnia, Fundación Universitaria Agraria de Colombia (UNIAGRARIA), Bogotá 111321, Colombia

[6] Thad Cochran National Warmwater Aquaculture Center, Mississippi Agriculture and Forestry Experiment Station, Mississippi State University, Stoneville, MS 38776, USA

[7] Department of Wildlife, Fisheries and Aquaculture, Mississippi State University, Starkville, MS 39762, USA

[8] Instituto Nacional de Pesquisas da Amazônia, Manaus 69067-375, AM, Brazil

* Correspondence: fyy5@msstate.edu (F.Y.Y.); ligia.goncalves@inpa.gov.br (L.U.G.)

**Abstract:** The black soldier fly (BSF, *Hermetia illucens*) is a prominent insect species and a novel alternative ingredient for aquafeeds. This study aimed to evaluate the replacement of commercial feed with fresh whole black soldier fly larvae (BSFL) for *Colossoma macropomum*. Juvenile tambaqui (115.2 ± 0.9 g/fish) were stocked in 800 L tanks (11 fish/tank) operating as a recirculating system. The dietary treatments consisted of whole larvae only (BSFL), 50:50 BSFL: commercial feed (BSFL: CF), or only commercial feed (CF), and the fish were fed to apparent satiation for 120 days. At the end of the feeding trial, fish were group weighed, and blood and muscle were collected for hematological and sensorial evaluations. Fish fed BSFL:CF presented with similar growth performance and carcass yield to tambaqui that were fed only CF. The high fat content of the larvae contributed to the highest values for the viscerosomatic index (7.01%; 6.56%), plasmatic triglycerides (312.1 mg/dL; 295.1 mg/dL), and cholesterol (120.6 mg/dL; 119.1 mg/dL) in the fish provided with BSFL or BSFL:CF, respectively. However, a better acceptance in the fillet sensory analysis was observed for these fillets than for the fillets from the CF group. Feeding tambaqui with BSFL:CF did not compromise the production performance and may have improved the acceptance of their fillets.

**Keywords:** aquafeeds; neotropical fish; insect larvae

## 1. Introduction

Feeding can represent up to 70% of the total operational costs for fed-aquaculture depending on the culture conditions, and the protein ingredients comprise the most expensive component in aquafeed formulations [1]. Insect meals are a novel protein source and a promising alternative feedstuff for monogastric farmed animals in the poultry and swine industries [2] as well as in aquaculture [3–4]. Meals manufactured from the insect larval phase can be a great protein source with a balanced amino acid profile

[5] and a rapid production turnover rate. Producing insect meals leaves a relatively low carbon footprint and makes more efficient use of land when compared with other agricultural practices, as measured by the volume of protein produced per area of land [6].

Black soldier fly larvae (BSFL), *Hermetia illucens*, is considered a prospective insect species for ingredient manufacturing and animal feeding, given the high protein content (37% to 63%) and how it presents with an amino acid composition comparable to fishmeal [4,7]. BSFL meal has been evaluated as an alternative feedstuff for the feeding of several fish species, such as the Atlantic salmon (*Salmo salar*), rainbow trout (*Oncorhynchus mykiss*), and Nile tilapia (*Oreochromis niloticus*), with successful studies reporting a 50 to 100% replacement of fishmeal [8–10].

Several insect production systems are currently being developed worldwide, ranging from those used by small-scale farmers to those suitable for multi-million-dollar facilities producing BSFL. Industrial production focuses on manufacturing the insect larvae, which has dried and ground meal or the defatted meal as the final product, yielding an ingredient with a higher protein concentration. Nevertheless, scientists and insect producers have been exploring the feasibility of using fresh whole insect larvae to feed farmed monogastric animals, with successful cases reported in African countries [11] and for whiteleg shrimp (*Litopenaeus vannamei*) [12]. Nevertheless, it should be highlighted that the practice of feeding fresh whole insects to animals may be restricted in some countries, such as those in the European Union (EU), which follow their animal feeding regulations No. 2017/893.

Tambaqui (*Colossoma macropomum*) is the most farmed fish species for food production and is endemic to the Brazilian region; it is a valuable commodity for the population residing in the Rain Forest region. In 2019, the production level for farmed Brazilian tambaqui reached a remarkable 101,079 tons, ranking second for Brazilian national farmed fish after the Nile tilapia. The *Colossoma macropomum* is also heavily cultured in other countries in South America, but production has also been reported in Central America and some Asian countries [13]. In their natural habitat, the tambaqui display omnivorous feeding behavior. However, during the drought season, their feeding behavior can change to a higher trophic level. Hence, when zooplankton and other arthropods are more readily available, this fish can start preying on these animals, which ultimately favors increased food diversification and a higher degree of omnivory [13].

The use of whole insect larvae in animal feed can be an interesting approach to lessening food waste and promoting a more efficient circular economy. This approach not only bypasses the manufacturing processes that require heavy machinery, but it also allows for a business model that is more inclusive, with small-hold farmers producing insect larvae from their food scraps and directly feeding the produced insect larvae to their fish. This practice can improve conditions by addressing social-economic and environmental issues in developing countries. In addition, the COVID-19 pandemic [14] and the conflicts in European countries have greatly disrupted the supply chains, with rising prices and a decreased availability of the ingredients commonly used in aquafeeds [15]. This economic instability has had a greater impact on small-scale farmers because of their limited resources. In this context, providing whole BSFL could be an appealing alternative for these producers, who may not be able to access or afford to manufacture extruded feeds year-round. Feeding whole BSFL to farmed fish can be an alternate strategy to not only recycle nutrients that are wasted but also to ensure food security and improve the production process of farmed fish for human consumption in developing countries.

Moreover, recycling nutrients from organic waste streams and improving their nutritional value for use as a novel feed ingredient for aquaculture can contribute to achieving the Sustainable Development Goals (SDG) 1, 2, 9, 12, 14, and 15, which were established by the United Nations for 2030 [16–17]. The objective of this study was two-fold: firstly, to evaluate the effects of replacing commercial feed (CF) with whole BSFL for

tambaqui, *Colossoma macropomum*, on growth performance and hematological parameters; secondly, to evaluate the fillets produced in this feeding trial for consumers' perceptions and acceptability during a sensory panel.

## 2. Materials and Methods

This study was carried out at the Aquaculture Experimental Station of the Coordination of Technology and Innovation (COTEI) of the National Institute of Amazon Research (INPA), Manaus, Amazonas, Brazil.

### 2.1. Feeding Trial

One hundred and thirty-two juvenile tambaqui (115.2 ± 0.9 g; average ± standard deviation) were evenly distributed in twelve 800 L tanks (11 fish per tank) operating as a recirculating system. The dietary treatments were distributed in a completely randomized experimental design, having 4 replicates per experimental group($n = 4$). Fish were fed until apparent satiation twice daily (08:00 and 16:00) for 120 days. The dietary treatments consisted of: only whole BSFL; 50% BSFL; 50% CF (BSFL:CF); only CF (CF, as the control group). For the BSFL:CF group, CF was supplied in the morning meal and the equivalent weight of whole BSFL was provided in the afternoon meal. The proximate composition and gross energy content of the commercial feed and whole black soldier fly larvae were analyzed with established procedures according to AOAC [18], with minor modifications (Table 1). Briefly, the estimated protein nitrogen from the BSFL was adjusted to a 5.7 factor to account for the chitin nitrogen from the larva exoskeleton. Frozen BSFL was obtained by the Entomology and Acarology Laboratory from the Federal University of Amazonas (FCA-UFAM) (Manaus, Brazil). During the larvae growth out stage, the BSFL was provided broiler feed as a rearing substrate (21% crude protein, 2.5% lipid; 5% crude fiber; as-is). The commercial broiler feed was given to the larvae to provide a standardized nutritional substrate to the BSFL throughout the tambaqui feeding trial. On day 14, the BSFL was sifted from the rearing substrate at an optimum weight and stored frozen at −20 °C. Before feeding the tambaqui, frozen larvae were allowed to thaw at room temperature for 1h prior to feeding. The procedures for this study were in compliance with the Ethical Committee of Research on the Use of Animals of the INPA (no. 056/2017) and the Human Research Ethics Committees of the UFAM (no. 4.288.497).

**Table 1.** Proximate composition and gross energy content of commercial feed (CF) and whole black soldier fly larvae (BSFL). Data are presented in a dry-matter basis.

| Proximate Composition and Energy | BSFL | CF |
|---|---|---|
| Dry Matter (g/kg) | 345.5 | 922.0 |
| Crude protein (g/kg) | 415.7 | 332.0 |
| Lipid (g/kg) | 238.5 | 39.0 |
| Crude fiber (g/kg) | 19.1 | 60.0 |
| Ash (g/kg) | 109.0 | 80.0 |
| Gross energy (MJ/kg) * | 23.7 | 17.3 |

Abbreviations: BSFL, Black soldier fly larvae; CF, Commercial Feed; *· Calculated energy = (5.7 × Crude protein) + (9.4 × Lipid) + [4.1 × (Non-nitrogenous extract + Crude fiber)]; Non-nitrogenous extract = 100 − (moisture + Crude protein + Lipid + Crude fiber + Ash).

### 2.2. Water Quality

Water quality parameters were monitored throughout the trial. Water temperature and dissolved oxygen levels were recorded weekly using an optical dissolved oxygen meter (ProODO, YSI Inc., Yellow Springs, OH, USA), and pH was measured using a digital pH meter (pH 100, YSI Inc.). The total dissolved ammonia, nitrite, and nitrate were measured twice a week with colorimetric kits (Alfakit AT 101; Alfakit, Florianópolis, Santa Catarina, Brazil). Water from the phytoremediation recirculation system conferred all the

evaluated water parameters stability, and the total ammonia and nitrite levels were within a suitable range for the production of tambaqui [13].

### 2.3. Growth Performance

During the feeding trial, fish mortality and feed intake were recorded daily. For the growth performance evaluation, fish were made to fast for 12 h prior to handling, anesthetized with clove oil (0.05 mL/L), and group weighed. The data collected were used to compute the production performance [19], calculated as follows:

Daily weight gain (g) = average weight gain ÷ period in days;

Daily Feed Intake (g) = total weight of feed consumed per fish/120 days;

Specific growth rate (%) = (ln (final weight)- ln (initial weight))/days;

Feed conversion rate = feed intake ÷ weight gain;

Daily protein intake = (Total Feed Intake × protein content (feed and/or larvae))/120 days;

Daily energy intake = (Total Feed Intake × energy content (feed and/or larvae))/120 days.

### 2.4. Hematological Analysis

At the end of the feeding trial, fish fasted for 12 h, and three fish from each experimental unit were randomly selected for blood collection and anesthetized using the aforementioned procedures. Blood samples from each fish were collected from the caudal vasculature using syringes coated with ethylenediaminetetraacetic acid (EDTA-10%). Whole blood samples were used for the hematocrit [20] and hemoglobin [Hb] [21] percentage analyses. Circulating erythrocyte (RBC) was measured using a Neubauer chamber in diluted blood (1:200) [22]. The hematimetric parameter of the mean corpuscular volume (MCV) was determined following the methods proposed by Wintrobe et al. [23]. The glucose was determined by the glucose oxidase method and the total protein by the biuret method with a commercial kit (Labtest, Lagoa Santa, Minas Gerais, Brazil). Cholesterol and triglycerides were established by the lipid-bleaching method using commercial kits (InVitro, MG, Brazil and VIDA Biotecnologia, MG, Brazil). Whole blood samples were centrifuged at 2200× $g$ for 3 min at 4 °C, and aliquoted plasma samples were used to measure the albumin concentration using a commercial kit (Labtest, Lagoa Santa, MG, Brazil). Analyses were performed in a Spectramax M5 (Molecular Device Inc., San Jose, CA, USA).

### 2.5. Body Analysis and Somatic Indices

Another subset of three fish per tank was euthanized using an overdose of clove oil (0.4 mL/L) followed by rinsing the fish with chlorinated water at 5 ppm for filleting and evisceration. Individual fish were weighed, and their respective viscera and fillet were computed to evaluate the somatic indices, fillet yield (skinned and deboned fillet), skin yield (covering the fish body), and head yield (sectioned from the body at the junction with the spine, including gills). The calculations used for the condition indices and yields were as follows:

Viserosomatic index = 100 × (visceras weight/body weight);

Hepatosomatic index = 100 × (liver weight/body weight);

Visceral fat index = 100 × (visceral fat weight/body weight);

Fillet yield = 100 × (Fillet weight/body weight);

Head yield = 100 × (Head weight/body weight);

Skin yield = 100 × (Skin weight/body weight).

### 2.6. Sensorial Analysis

For the sensorial panel of fish fillets, four fish per experimental unit were immersed in ice water (3 °C) to decrease metabolism and muscle reflexes. The fish were then

euthanized by rupture of the spinal cord. Samples of fillets were fractioned in 3 g portions and stored at −20 °C prior to the sensorial evaluations. Fillet samples were subjected to *Salmonella* spp., coagulase-positive *Staphylococcus* spp., and total coliforms tests in the Microbiology Laboratory from Nilton Lins University. The microbiological results were within the range for food safety and suitability for human consumption required by the Brazilian National Health Surveillance Agency (ANVISA).

Fish fillets were steam cooked for 15 min and evaluated for color, aroma, flavor, texture, odor, appearance, and general acceptance by an untrained test panel of 55 volunteers. Fillets from all treatments were blindly scored by each volunteer. The 9 point hedonic scale, ranging from 1 = dislike intensely to 9 = like intensely, with a value of 5 signifying neither like nor dislike, was used in the evaluation [24]. The volunteer panelists gave their written consent for this test. This sensorial panel was approved by the ethics committee for research with humans of the Federal University of Amazona (Protocol#: 4.288.497), and all participants completed a consent form outlining the confidentiality and de-identification of the collected data.

### 2.7. Statistical Analysis

The resulting data for production performance, condition indices, yield, and hematological variables were subjected to a one-way ANOVA. The assumption of the homogeneity of variances was verified by Levene's test, and the normality of residuals was assessed by the Shapiro–Wilk test. When results were found to be significant ($p < 0.05$), the data were further subjected to a comparison of means using the Tukey HSD test. The sampling error (variation among the fish in the same tank) was included in the model as a random effect on the model. The sensorial analysis data were analyzed by Friedman's nonparametric test, which evaluates the effect of treatments considering the effect of the judge. If significance was met ($p < 0.05$), a post hoc Nemenyi test adjusted to $\chi^2$ statistics was performed for the multiple pairwise comparisons. The R project program carried out the data analysis (R Development Core Team, 2019).

## 3. Results

The average values for the water quality parameters throughout the trial were as follows: water temperature (27.8 ± 0.71 °C, average ± standard deviation, SD), dissolved oxygen (7.84 ± 0.77 mg/L), pH (6.49 ± 1.01), total ammonia (0.46 ± 0.42 mg/L), nitrite (0.1 ± 0.11 mg/L), and nitrate (0.94 ± 0.62 mg/L). The survival rate during the experimental period was 100%, and no apparent rejection of feed pellets and BSFL was recorded. Fish fed BSFL:CF presented similar body weight with fish fed CF only at 68, 98, and 120 days. The daily weight gain, specific growth rate, and feed conversion rate for fish fed BSFL:CF were also not statistically different when compared to the CF-treated fish. Fish fed BSFL presented lower production performance for all parameters but an improved feed conversion rate (Table 2).

**Table 2.** Growth performance (average ± SEM) of the tambaqui fed whole black soldier fly larvae (BSFL) to replace commercial feed (CF).

| Variables | BSFL | BSFL:CF | CF | *p*-Value |
|---|---|---|---|---|
| Body weight (g) 36 days | 164.92 ± 7.74 | 179.11 ± 5.13 | 186.18 ± 11.01 | 0.237 |
| Body weight (g) 68 days | 213.65 ± 9.18 b | 239.33 ± 5.89 ab | 263.33 ± 8.66 a | 0.006 |
| Body weight (g) 98 days | 247.30 ± 7.85 b | 306.85 ± 8.28 a | 320.49 ± 14.24 a | 0.002 |
| Final body weight (g) | 285.92 ± 11.69 b | 337.96 ± 11.04 ab | 364.67 ± 16.67 a | 0.007 |
| Daily weight gain (g) | 1.40 ± 0.10 b | 1.83 ± 0.09 ab | 2.04 ± 0.14 a | 0.007 |
| Specific growth rate (%) | 0.74 ± 0.03 b | 0.88 ± 0.03 a | 0.94 ± 0.04 a | 0.006 |
| Daily feed intake (g, dry matter) | 1.80 ± 0.04 c | 2.82 ± 0.10 b | 3.88 ± 0.09 a | 0.000 |
| Feed conversion rate | 1.31 ± 0.08 b | 1.55 ± 0.07 ab | 1.92 ± 0.12 a | 0.004 |
| Daily protein intake (g/day) | 0.75 ± 0.04 c | 1.17 ± 0.10 b | 1.61 ± 0.09 a | 0.001 |
| Daily energy intake (MJ/day) | 0.04 ± 0.00 c | 0.05 ± 0.00 b | 0.07 ± 0.00 a | 0.001 |

Mean values with different letters on the same line were significantly different according to the Tukey test ($p < 0.05$).

Significant differences were also observed for the feed intake where the fish fed CF presented the highest feed intake followed by the fish fed BSFL:CF, and the CF treated group presented the lowest feed intake. Daily protein intake and energy intake were also significantly impacted by the dietary treatments in the same fashion as the feed intake. No significant differences were detected for fillet yield, head yield, and the hepatosomatic index. However, fish fed BSFL presented higher viscerosomatic index values when compared to tambaqui fed CF, with no detectable differences for fish fed BSFL:CF (Table 3).

**Table 3.** Corporal indices and fillet yield (average ± SEM) of the tambaqui fed whole black soldier fly larvae (BSFL) to replace commercial feed (CF).

| Variables | BSFL | BSFL:CF | CF | *p*-Value |
|---|---|---|---|---|
| Viscerosomatic index (%) | 7.01 ± 0.15 a | 6.56 ± 0.16 ab | 6.20 ± 0.20 b | 0.025 |
| Hepatosomatic index (%) | 1.40 ± 0.05 | 1.50 ± 0.07 | 1.61 ± 0.09 | 0.161 |
| Visceral fat index (%) | 2.79 ± 0.33 | 2.36 ± 0.28 | 2.19 ± 0.12 | 0.282 |
| Fillet (%) | 28.04 ± 0.58 | 28.49 ± 0.37 | 28.09 ± 0.25 | 0.719 |
| Head (%) | 19.63 ± 0.85 | 19.78 ± 0.97 | 18.44 ± 0.41 | 0.539 |
| Skin (%) | 5.47 ± 0.28 | 5.58 ± 0.22 | 5.51 ± 0.34 | 0.967 |

Mean values with different letters on the same line were significantly different according to the Tukey test ($p < 0.05$).

The dietary treatments did not influence hematocrit, mean corpuscular volume, red blood cells, and plasmatic glucose and protein. The plasma triglycerides and cholesterol levels significantly increased ($p < 0.05$) for the fish fed BSFL:CF and BSFL (Table 4).

**Table 4.** Hematological and biochemical parameters (average ± SEM) of the tambaqui fed whole black soldier fly larvae (BSFL) to replace commercial feed (CF).

| Parameter | BSFL | BSFL:CF | CF | *p*-Value |
|---|---|---|---|---|
| Hematocrit (%) * | 33.1 ± 4.1 | 38.8 ± 0.7 | 37.2 ± 2.3 | 0.44 |
| Mean corpuscular volume (fL) | 145.6 ± 16.3 | 167.7 ± 10.1 | 193.5 ± 19.0 | 0.15 |
| Red blood cells ($10^6$/μL) | 2.34 ± 0.18 | 2.38 ± 0.14 | 2.09 ± 0.26 | 0.58 |
| Plasma glucose (mg/dL) | 138.9 ± 13.7 | 112.9 ± 11.7 | 104.7 ± 12.3 | 0.19 |
| Plasma triglycerides (mg/dL) | 312.1 ± 9.6 a | 295.1 ± 12.2 a | 200.2 ± 15.0 b | 0.001 |
| Total plasma protein (g/dL)* | 4.09 ± 0.06 | 4.15 ± 0.57 | 3.87 ± 0.22 | 0.38 |
| Plasma cholesterol (mg/dL) | 120.6 ± 6.1 a | 119.1 ± 17.3 a | 83.4 ± 3.1 b | 0.001 |

Mean values with different letters on the same line were significantly different according to the Tukey test ($p < 0.05$). *. Hematocrit and total plasma protein were analyzed by the Kruskal–Wallis test.

The dietary treatments influenced neither the color nor the texture of the fillets. The flavor, odor, and appearance attributes were better graded for the fillets of fish that were fed BSFL:CF than for those from the CF fish. Nonetheless, these fish did not differ from the fish fed BSFL only (Table 5).

**Table 5.** Sensorial attributes of the fish fillet quality (average ± SEM) of the tambaqui fed whole black soldier fly larvae (BSFL) to replace commercial feed (CF).

| Atributes | BSFL | BSFL:CF | CF | *p*-Value |
|---|---|---|---|---|
| Color | 7.29 ± 0.18 | 7.47 ± 0.22 | 6.78 ± 0.26 | 0.180 |
| Texture | 7.80 ± 0.21 | 7.98 ± 0.15 | 7.60 ± 0.17 | 0.072 |
| Flavor | 7.58 ± 0.18 ab | 7.73 ± 0.19 a | 7.00 ± 0.20 b | 0.003 |
| Odor | 7.04 ± 0.22 ab | 7.33 ± 0.23 a | 6.65 ± 0.27 b | 0.014 |
| Apparence | 7.16 ± 0.25 ab | 7.60 ± 0.23 a | 6.60 ± 0.29 b | 0.01 |

Mean values with different letters on the same line were significantly different according to Friedman's nonparametric test ($p < 0.05$).

## 4. Discussion

There is an urgent and widespread concern to reduce the amount of food waste generated by households, groceries, industries, and governmental sectors [25]. In order to properly reutilize this surplus of nonhuman-grade food, there are several research efforts aiming to recycle these waste streams into useful nutrients for feeding animals or for composting [26]. For instance, restaurant post-consumer food waste [27], grains unfit for human consumption [28], and fresh BSFL [12] are a few examples of prospective organic waste that can partially replace commercial feed in aquaculture. In this context, feeding black soldier fly larvae straight to the fish can be a promising alternative, because the nutritional makeup of the larvae contains high amounts of protein and energy. In our study, tambaqui juveniles readily accepted the dietary whole BSFL, and at no moment was there any rejection by the fish.

The production parameters observed for daily weight gain, specific growth rate, and daily feed intake were similar to the fish that received the BSFL:CF and CF diets, but the variable feed intake differed between the diets. This could be an indication that the consumption had a direct impact on the outcome of productive performance. Feed intake could have been influenced by the apparent satiation owing to the consumed volume of whole BSFL, which can take up more volume in the stomach because of its high moisture content, or by the high lipid content of BSFL [19].

The high lipid content of the BSFL increases the gross dietary energy of feeding, and high-energy diets are known to achieve early satiation in animals, which could have limited the intake of the fish treated with whole BSFL [19,29]. However, the combination of BSFL:CF proved to be sufficient to ensure that the required nutrients and energy were achieved for growth, with comparable values presented by the tambaqui treated with CF. The great ability of neotropical freshwater fish to digest lipids [30], and possibly protein-sparing effects due to high levels of energy [31–32], may have improved the feed conversion of fish treated with BSFL:CF and BSFL. On the other hand, the daily protein and energy intake were significantly higher than for the fish fed commercial feed only, suggesting that perhaps the high moisture content provided by the BSFL may be more attractive to the fish, allowing them to ingest more nutrients.

The dietary treatments did not influence circulating glucose in the plasma of tambaqui. Glucose synthesis from lactate, amino acids, and glycerol has been indicated in teleost fish [33]. For the catfish *Rhamdia quelen*, changes in the activity of gluconeogenic enzymes as a response to dietary protein content have been reported [34]. In the present study, the plasma glucose averaged 105 mg/dL for the fish fed CF and 139 mg/dL for those fed BSFL, with no statistical differences observed. On the other hand, these values are similar to those from previous studies for this species, which ranged from 101 mg/dL to 120 mg/dL [35–36].

In fish, the excess dietary lipid is exported from the liver in the form of lipoproteins (very low-density lipoprotein) to be stored in specific storage sites [37]. The lipid content of BSFL directly influenced the concentration of triglycerides in the plasma of tambaqui, as observed in the triglyceride levels in the dietary group treated with BSFL only. Conversely, studies have related a decrease in triglycerides in the plasma of fish fed diets with BSF as a substitution for fish meal [38], which can be related to dietary chitin occurrence and its potential to bind bile acids and free fatty acids [39]. However, in the present study, tambaqui juveniles that were offered BSFL only developed the ability to peel the insect larvae and consume only the internal contents, leaving the external structure (exoskeleton) behind. This feeding behavior may have reduced the chitin intake, which could have increased the plasma triglyceride levels. Nevertheless, the plasma triglyceride values for tambaqui observed in the present study were within the normal ranges indicated in the literature for this species [40]. Tambaqui fed on commercial feed or that are on diets with

different levels of carbohydrates and lipids presented triglyceride levels ranging from 196.61 mg/dL to 458.84 mg/dL [41], values which are somewhat similar to those found in this study.

The plasma total cholesterol levels in tambaqui were significantly affected by offering the BSFL:CF and BSFL treatments. Interestingly, cholesterol levels were lower in the tambaqui fed the CF diet, and it could be related to the plant-based ingredients in the formula of CF. Geay et al. [42] suggested a lipogenic pathways stimulation in European sea bass (*Dicentrarchus labrax*) fed a plant-based diet due to the up-regulation of several genes involved in the long-chain polyunsaturated fatty acids and cholesterol biosynthetic pathway. This hypocholesterolemic effect was also observed in gibel carp (*Carassius gibelio*) [43] and turbot (*Scophthalmus maximus*) fed plant-based diets [44]. For tambaqui, the variation in plasma cholesterol concentrations can range from 59.64 to 104.84 mg/dL [36], and the present findings show that the tambaqui's cholesterol may have been altered by the concentration of the fatty acid profile which is available in the food. Although changes were observed in the cholesterol levels of the fish treated with BSFL, the presented parameters were within a normal range for this species [36].

The increased viscerosomatic index of the fish fed BSFL can also be directly related to the high lipid content of BSFL. Previous studies with several fish species have shown an increased viscerosomatic index with increasing dietary fat [45]. In the same fashion, the increase in plasma triglycerides and cholesterol in the fish fed BSFL were likely related to the high lipid content of BSFL. Similar results were observed for Nile tilapia, which presented increased plasma levels of these lipid compounds in response to rising dietary fat levels [46]. However, an alternative hypothesis to explain the higher viscerosomatic index of the BSF-fed fish is their lower body weight when compared to the CF-treated fish.

Interestingly, adding BSFL to the tambaqui diet improved the flavor, odor, and appearance of the fish fillets, especially for the fillet sampled from the fish fed BSFL:CF. In line with the present findings, a previous study from Borgogno et al. [47] evaluated the inclusion of black soldier fly larvae meal with rainbow trout and observed that 12 of 19 sensorial attributes were influenced by diets that included BSFL meal. However, another study with rainbow trout reported that the replacement of 50% dietary fish meal with BSFL meal did not influence the fillets' sensory properties and acceptance judgements [9]. Changes in fillet flavor perception could also be a response to the modifications in the lipid content and the fatty acid profile caused by the increased use of BSFL in the fish feed [47]. More studies investigating the consumers' acceptance of fish fed BSFL and their correlation between the amino acid and fatty acid profile are suggested.

It is noteworthy to highlight that the fish fed BSFL:CF achieved comparable growth to those strictly fed CF, which would be considered ideal production values for commercial fish farming. On the other hand, fish fed whole BSFL presented a higher fillet grade for sensory quality, which can bring a premium value to the final fillet product. Therefore, this preliminary study showed that the use of whole BSFL as part of fish feeding is a feasible approach and can assist with the familiar fish farming developments of rural families, which will ultimately contribute to the development of the rural social-economic and sustainable production of farmed fish. Future studies should pinpoint the causes of the possible nutritional imbalances when providing BSFL only—such as the high energy to protein ratio, high fat to digestible carbohydrates ratio, or vitamin deficiencies—and perform an economic evaluation to measure the viability of rearing BSFL to replace 50% of the commercial pellets.

## 5. Conclusions

Feeding tambaqui juveniles with BSFL:CF for 120 days kept the blood parameters of the fish within acceptable ranges and did not compromise their growth performance. The use of whole BSFL also provided a better sensory quality compared to fish raised only with CF. Whole black soldier fly larvae can replace half of the dietary extruded feed for juvenile tambaqui.

**Author Contributions:** Conceptualization, L.U.G., B.M.O., T.M.S. and D.P.C.; methodology, B.M.O., D.P.C. and L.U.G.; validation, B.M.O., D.P.C., L.C.C.M. and T.M.S.; formal analysis, L.U.G., F.A.A.A., G.A.P.P. and F.Y.Y.; investigation, all authors; resources, L.U.G. and N.P.T.F.; data curation, B.M.O., T.M.S., D.K.M.d.S., F.A.A.A. and L.U.G.; writing—original draft preparation, B.M.O., L.C.C.M., D.P.C. and F.A.A.A.; writing—review and editing, T.M.S., D.K.M.d.S., F.Y.Y., G.A.P.P. and L.U.G.; visualization, B.M.O., L.U.G. and F.A.A.A.; supervision, L.U.G. and F.A.A.A.; project administration, L.U.G., and F.A.A.A.; funding acquisition, L.U.G. and N.P.T.F. All authors have read and agreed to the published version of the manuscript.

**Funding:** This study was sponsored by the National Institute of Amazonian Research (INPA); Amazonas State Research Foundation (FAPEAM 008/2021,); Coordination for the Improvement of Higher Education Personnel (CAPES); Council for Scientific and Technological Development (CNPq 312492/2021-9).

**Institutional Review Board Statement:** The procedures of this study were in compliance with the Ethical Committee of Research on the Use of Animals of the INPA (no. 056/2017) and the Human Research Ethics Committees of the UFAM (no. 4.288.497).

**Informed Consent Statement:** Informed consent was obtained from all subjects involved in the study.

**Data Availability Statement:** Data from the study is available from the corresponding authors upon reasonable request.

**Acknowledgments:** The authors gratefully acknowledge Amazonas State Research Foundation (FAPEAM) for the research grant (008/2021—PROSPAM/FAPEAM). D.P.C. received a master fellowship from Coordination for the Improvement of Higher Education Personnel—CAPES. T.M.S. is a doctoral scholarship from CAPES. D.K.M.d.S. received a doctoral scholarship from FAPEAM; L.U.G is a research fellow from the Brazilian National Council for Scientific and Technological Development CNPq (Process 312492/2021-9). The authors would like to thank the staff of the National Institute of Amazonian Research (INPA) and the students who assisted in this project.

**Conflicts of Interest:** The authors declare there are no conflict of interest.

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
