# Peer review of "Whole Black Soldier Fly Larvae (Hermetia illucens) as Dietary Replacement of Extruded Feed for Tambaqui (Colossoma macropomum) Juveniles"

_2673-9496, doi:10.3390/aquacj2040014_

Round 1
Reviewer 1 Report (Previous Reviewer 2)
Just make the few corrections suggested with little modification of the topic.
Use the more conventional method of representing proximate analysis by including the percentage composition in parenthesis ( ) in Tabe 1.

Author Response
REVIEWER #1
- The topic may sound better by recaptioning it as: Whole black soldier fly larvae (Hermettie illucens) as dietary replacement of extruded feed for tambaqui (Colossoma macropomum) juvenile
Thank you for your suggestion. The title was modified as requested.
- L.24. Th replace with The
Thank you for your suggestion. The typo was fixed.
L.25. delete 'This insect can recycle...streams'. or move to introduction
Thank you for your suggestion. Modifications were made as requested.
L.33 & L.34 insert the high values in parenthesis for better summary.
Thank you for your suggestion. Results were included in the abstract.
L.119 The BSFL changes to 'the' BSFL
Thank you for noticing. The typo was fixed.
L.121 delete 'the' before nutritional
Thank you for your suggestion. Modifications were made to better deliver the information.
- L.128-L.130 i.e Table 1. Whether a factor of 6.25 or 5.25 was used during computation does not still change the values as Centesimal presentations as such. It is recommended that the percentage be put in parenthesis e.g CP 415.7 (41.57%) and so, for easier readability and comprehension to a wider audience.
With the correction of these additional inputs in place, proceed to publish.
Thank you for your suggestions. The units expressed follows the guidelines provided by the Aquaculture Journal. Thus, the authors would like to keep the presentation of the results as-is.

Reviewer 2 Report (New Reviewer)
Dear Author(s),
In the present study, Ordonez et al. addressed the Whole black soldier fly larvae (Hermetia illucens) can replace half of the dietary extruded feed for juvenile tambaqui (Colossoma macropomum)
- In general, the aim of the study is clearly defined. The study is a survey experiment. The authors are approaching the results only by comparing the means obtained with each specific treatment or comparing to previous publications. However, some important concerns need to arise from this study, particularly those regarding novelty and discussion.
- Line 24, replace ‘th’ with ‘the’
- Regrading the experimental design for BSFL:CF treatment group, authors states that “CF was supplied in the morning meal and the equivalent weight of whole BSFL was provided in the afternoon meal”. Is there any impact in case of separation fishmeal?
- Why the authors selected the replacement fishmeal at the doses of 0%, 50%, and 100%?
- What kind of commercial feed was used in this study? Please clarify.
- The unit for clove oil (Line 145 and 171) should be consistent.
- Line 222-223, please review the description again. It does not make sense.
- Line 308-309, authors state that “although changes were observed in cholesterol levels in the fish fed with BSFL, the presented parameters were still within normal range for this species”. What is normal range of tambaqui? Did authors rely on their own conclusions or other sources to establish this issue?
- Line 332, ‘fed’ word is duplicate.
- The authors identified a number of limitations in each paragraph of discussion section and they should be addressed in future studies, causing the decreased quality of this research. Please consider.
Author Response
REVIEWER #2
In the present study, Ordonez et al. addressed the Whole black soldier fly larvae (Hermetia illucens) can replace half of the dietary extruded feed for juvenile tambaqui (Colossoma macropomum)
- In general, the aim of the study is clearly defined. The study is a survey experiment. The authors are approaching the results only by comparing the means obtained with each specific treatment or comparing to previous publications. However, some important concerns need to arise from this study, particularly those regarding novelty and discussion.
Thank you for your comments and questions. We hope that we have addressed your concerns and we remain at your disposal for further clarification.
- Line 24, replace ‘th’ with ‘the’
Thank you for your suggestion. The typo was fixed.
- Regrading the experimental design for BSFL:CF treatment group, authors states that “CF was supplied in the morning meal and the equivalent weight of whole BSFL was provided in the afternoon meal”. Is there any impact in case of separation fishmeal?
CF is the abbreviation of commercial feed. This feed were manufactured through the extrusion of a mixture of ingredients. There is no fishmeal evaluation in our study.
- Why the authors selected the replacement fishmeal at the doses of 0%, 50%, and 100%?
Dietary fishmeal was not evaluated in this study. Commercial extruded feed was replaced by fresh black soldier fly larvae at 0, 50, and 100%. These levels were selected from the number of fish and experimental units available at the time, and volume of black soldier fly larvae produced per day.
- What kind of commercial feed was used in this study? Please clarify.
The commercial extruded feed for Neotropical omnivore fish (Multifós®), with pellet size of 6-8 mm. The nutritional value of the feed is described in table 1.
- The unit for clove oil (Line 145 and 171) should be consistent.
Thank you for your suggestion. Modifications were made as requested.
- Line 222-223, please review the description again. It does not make sense.
Thank you for your observation. The sentence was revised.
- Line 308-309, authors state that “although changes were observed in cholesterol levels in the fish fed with BSFL, the presented parameters were still within normal range for this species”. What is normal range of tambaqui? Did authors rely on their own conclusions or other sources to establish this issue?
Thank you for your comment. The cholesterol range is mentioned in the previous sentence with the literature cited.
- Line 332, ‘fed’ word is duplicate.
Thank you for your input. The extra word was deleted.
- The authors identified a number of limitations in each paragraph of discussion section and they should be addressed in future studies, causing the decreased quality of this research. Please consider.
The authors agree with the reviewer’s suggestion, and some of these recommendations were excluded.

Reviewer 3 Report (New Reviewer)
The idea of the work is interesting, and the experimental set is well described, however results, discussion and conclusions are careless and neglected. A re-edition and a more accurately and precise description and contextualisation of results is required.
Following some major consideration:
- General revision of English is required
- Grammar and misprints revision is required: for example capitol letter of “The” at line 119
- Revision of references. There are 44 in the text and only 43 listed. This is a big mistake, which point out a careless revision from authors. Moreover, please provide some recent references. The most you cited are from 2005 to 2015.
- The aim of the studied is not well explicit in the text but only in the abstract. Please clarify within the text, so that discussion and conclusion should be more cohesive. Moreover, insert in tables the same abbreviation you cite in the text.
- Due to the long list of abbreviation the paper is difficult and messed up to read. I suggest providing a list of abbreviation at the beginning of the work. For example: line 221 you talk about FCR (Table 2) but in table 2 you write “Feed conversion rate”. Please revise. Also Line 265 you talk about DFI which is not well explicit in the text before. I guess you were talking about daily feed intake (DFI). Please clarify.
Discussion
This paragraph required separate comments.
First of all, please check references. Due to the messed up of citation discussion paragraph is difficult (sometimes impossible to read).
L.265-267. Can you explain me this sentence please? It sounds in conflict with line 221-223
Line 271. You cited [29]: I expected there was a parallel on the use of BSFL as feed, can you explain better why you decide to cite this reference?
Line 284 & others: you talk about future study. This is ok but what about to look for other studies on BSFL in literature?
Line 286: you declare that dietary did not influence circulating glucose. What about the observable decreasing trend of glucose, triglyceride and cholesterol in BSFL and CF?
L.293: there is the citation of references as they talk about BSFL but they did not. Can you explicit why and how you compare your study with the literature ones?
L.296 when you talk about “this study”, which study do you refer to?
L. 298-300 chitin intake. Where chitin is cited in the reference you report in the text? Why do you compare two different families of fish? Why do you compare different fish families which are fed with different composition of feed? In these references, any species of larvae is mention. I expected to have a comment on this aspect. I understand you would like to compare different kind of aquafeed already treat in literature, but I will appreciate a more accurate in-depth analysis on BSFL use in aquafeed.
Line 302- Line 306 Please give the correct reference.
Generally all the discussion paragraph is required to be re-edit.
Conclusions: Inconclusive, please reformulate.
Author Response
The idea of the work is interesting, and the experimental set is well described, however results, discussion and conclusions are careless and neglected. A re-edition and a more accurately and precise description and contextualisation of results is required.
Thank you for your comments and questions. We hope that we have addressed all your concerns and we remain at your disposal for further clarification.
Following some major consideration:
General revision of English is required
Thank you for your comment. The manuscript was revised and edited to improve the manuscript’s readability. We hope that we have addressed this concern.
-Grammar and misprints revision is required: for example capitol letter of “The” at line 119
Thank you for your suggestion. The typo was fixed.
-Revision of references. There are 44 in the text and only 43 listed. This is a big mistake, which point out a careless revision from authors. Moreover, please provide some recent references. The most you cited are from 2005 to 2015.
Thank you for your comment. Indeed, the references were poorly listed, and we apologize for this mistake. This manuscript has been previously revised and during these revisions some references were forgotten/misplaced. On the subject of the reference list, over 50% of the references were studies from 2017 to 2022, and over 75% were published from 2011 to 2022. Only 15% of the references listed were from 1985 to 2011. Please find below a histogram that shows the frequency of the references by year. The four outliers are older references from the material and methods. (FIGURE ON THE DOCUMENT ATTACHED)
-The aim of the studied is not well explicit in the text but only in the abstract. Please clarify within the text, so that discussion and conclusion should be more cohesive. Moreover, insert in tables the same abbreviation you cite in the text.
The aim of the study is presented in the abstract as well as in the last paragraph of the introduction. We believe that the discussion and conclusion addressed the objectives proposed by our research group. As stated by the reviewer, there is a long list of abbreviations, and that can make the manuscript difficult to read. We reduced the abbreviations throughout the manuscript.
-Due to the long list of abbreviation the paper is difficult and messed up to read. I suggest providing a list of abbreviation at the beginning of the work. For example: line 221 you talk about FCR (Table 2) but in table 2 you write “Feed conversion rate”. Please revise. Also Line 265 you talk about DFI which is not well explicit in the text before. I guess you were talking about daily feed intake (DFI). Please clarify.
Thank you for your suggestion, the manuscript was revised accordingly. Yes, DFI is daily feed intake. As previously stated, the acronyms have been removed to clarify the text.
Discussion
This paragraph required separate comments.
First of all, please check references. Due to the messed up of citation discussion paragraph is difficult (sometimes impossible to read).
Thank you for your observation. The references were corrected.
L.265-267. Can you explain me this sentence please? It sounds in conflict with line 221-223
We apologize for this typo. The treatment we were mentioning was CF instead of BSFL, which conflicted with the L221-223. These sentences were revised accordingly.
Line 271. You cited [29]: I expected there was a parallel on the use of BSFL as feed, can you explain better why you decide to cite this reference?
The reference was changed for the NRC, which states that the increase of fat/energy lead to satiation. This was observed for our fish fed the high-fat BSFL.
Line 284 & others: you talk about future study. This is ok but what about to look for other studies on BSFL in literature?
Thank you for your suggestion. Discussing our results was cumbersome as just one study had a similar experimental approach. Moreover, there were several studies that referenced manufactured BSFL meals as replacements for fishmeal.
Line 286: you declare that dietary did not influence circulating glucose. What about the observable decreasing trend of glucose, triglyceride and cholesterol in BSFL and CF?
The authors refrained to speculate about the non significant results for plasma glucose. Moreover, the reduced triglyceride and cholesterol were better discussed in the manuscript.
L.293: there is the citation of references as they talk about BSFL but they did not. Can you explicit why and how you compare your study with the literature ones?
The references were meant to show the normal ranges of triglycerides for the species. To the authors' knowledge, no black soldier fly larvae study was previously published for this species.
L.296 when you talk about “this study”, which study do you refer to?
Thank you for your question. This sentence was modified for better clarification.
- 298-300 chitin intake. Where chitin is cited in the reference you report in the text? Why do you compare two different families of fish? Why do you compare different fish families which are fed with different composition of feed? In these references, any species of larvae is mention. I expected to have a comment on this aspect. I understand you would like to compare different kind of aquafeed already treat in literature, but I will appreciate a more accurate in-depth analysis on BSFL use in aquafeed.
The influence of chitin on the lipid metabolism was better discussed in the mansucript. Moreover, two differente species of fish were compared because up to date there are no other references evaluating BSFL for tambaqui. The other species were freshwater with similar feeding habit. As previously mentioned, it is cumbersome to discuss our data, as this was a different approach than a classic fishmeal replacement. Up to date, only one study had a similar approach, which was evaluating fresh black soldier fly larvae for Whiteleg shrimp.
Line 302- Line 306 Please give the correct reference.
The correct references were adjusted. The authors apologize for this confusion.
Generally all the discussion paragraph is required to be re-edit.
Conclusions: Inconclusive, please reformulate.
The conclusion was modified as requested.

Round 2
Reviewer 2 Report (New Reviewer)
Dear authors, thank you very much for your corrections below I leave my comments on the acceptance of the manuscript. I can see that you have made a great effort to correct and add the requested information.
Author Response
Thank you for your input and comments.
Reviewer 3 Report (New Reviewer)
I would like to thank the authors for their care revision of the manuscript.
Author Response
Thank you for your input and comments.
This manuscript is a resubmission of an earlier submission. The following is a list of the peer review reports and author responses from that submission.
Round 1
Reviewer 1 Report
Dear authors,
I have just finished to revise the manuscript entitled "Whole black soldier fly larvae (Hermetia illucens) can replace half of the dietary extruded feed without compromising the production performance of juvenile tambaqui Colossoma macropomum". Tha paper try to go ahead the research on alternative protein sources for aquaculture, one of the mail covered topic on the last years. The new approach proposed is undoubtely interesting. Indeed, the use of whole larvae instead of larva meal should cut down the prices and can be a way to really improved small-scale farms. Nevertheless, thi way of administering insects is not allowed, at leats in Europe, and the introduction should report this. In addition, such a zootechnical trial is limited by the fact that the diet are not "iso" at all. For this reason, comparison between the experimental diets is really hard and, in my opinion, the discussion should be considered the energy and nutrient intake more than the feed intake.
Another doubt is about feeding administration of BSF:CF diet. How did you calculate the overall daily feed intake? Did you notice a different intake between the two meals? In other words, the presented DFI derived from the equal consumption of CF and BSF or not?
Other minor comments and suggestions are shown in the attached file.
Overall, I think that the purpose is good and the trial is interesting but the results should be considered looking at the nutrient intake since diet are not isoproteic,isolipidic and isoenergetic. For these reasons, major revisions are needed.

Author Response
Reviewer #1
Dear authors,
I have just finished to revise the manuscript entitled "Whole black soldier fly larvae (Hermetia illucens) can replace half of the dietary extruded feed without compromising the production performance of juvenile tambaqui Colossoma macropomum". Tha paper try to go ahead the research on alternative protein sources for aquaculture, one of the mail covered topic on the last years. The new approach proposed is undoubtely interesting. Indeed, the use of whole larvae instead of larva meal should cut down the prices and can be a way to really improved small-scale farms. Nevertheless, this way of administering insects is not allowed, at leats in Europe, and the introduction should report this. In addition, such a zootechnical trial is limited by the fact that the diet are not "iso" at all. For this reason, comparison between the experimental diets is really hard and, in my opinion, the discussion should be considered the energy and nutrient intake more than the feed intake.
Thank you for your suggestion. A sentence was added regarding the lesgislation in the EU. Daily protein intake and daily energy intake were added in the results and discussion.
Another doubt is about feeding administration of BSF:CF diet. How did you calculate the overall daily feed intake? Did you notice a different intake between the two meals? In other words, the presented DFI derived from the equal consumption of CF and BSF or not?
The feed intake was measured in the morning, and during the afternoon, a ration with equivalent weight would be prepared to feed the animals during the afternoon.
Other minor comments and suggestions are shown in the attached file.
Overall, I think that the purpose is good and the trial is interesting but the results should be considered looking at the nutrient intake since diet are not isoproteic,isolipidic and isoenergetic. For these reasons, major revisions are needed.
Thank you for your comment. We hope to have addressed your concerns.
L242: also chitin...have you considered this hyp? Because the FCR is better for BSF group than for the others despite the growth is partially inhibited. Furthermore, diets are not iso hence it is difficult to compare growth performace.
We did not take to account the chitin fraction because the group fed BSF only presented a higher FCR than the other dietary treatments. Thus, the observed results were linked to the high energy level provided by the lipid content from the BSF.
L260: This is really interesting! did you record it in some ways?
Unfortunately, this was not recorded nor graded in a way for us to report. It was a consistent visual observation.
L284-287: There are several work about sensory and lconsumer liking in species as salmon, trout ecc...it should be more appropriate to mention these ones and avoid laying hens.. do you mean that glutammic acid is not digested by directly absorbed ans stored in muscle? do you mean that glutammic acid is not digested by directly absorbed ans stored in muscle?
Thank you for your comment. We decided to delete this segment and replace with a sentence referencing a study conducted with rainbow trout. After getting a better understanding of the amino acid metabolism, we learned that a higher gluatmmic acid ingestion could not be directly related a higher glutammic acid in the fillets (Li et al., 2020). In addition, because we did not have the amino acid profile of fillets, we decided to discard this hypothesis.
Li et al., 2020. Nutrition and metabolism of glutamate and glutamine in fish. Amino Acids, 52:671-691. https://doi.org/10.1007/s00726-020-02851-2
L293-303: In my opionion this section should subsitute the conclusion. I'd rather this part more than the present conclusions.
The authors would like to keep this paragraph as is. It pertains useful information, and builds up for the conclusion of the manuscript.

Reviewer 2 Report
Recalculate the parameters in tables 1 and 2.

Author Response
Reviewer #2
L.27 juveniles of tambaqui (115.2g) stocked in 800-l tanks (11 fish/tank). Recast. Does 115.2g represent the total weight of the 11 fish/tank or average initial weight per individual? If so, indicate that. Let us know the total population stocked and the number of tanks used.
Thank you for your correction. The text was modified accordingly.
L.29 – delete BSF i.e larvae only (BSF),
Thank you for your correction. The text was modified accordingly.
L.30, 31 group were weighed Please indicate final weight.
- Result of hematological and sensorial should be presented
- 32- Fish fed BSF:CF presented similar growth performance ( ) and carcass yield ( ) to tambaqui fed CF ( ).
- 33- The fat content (23.85%) of BSF does not agree with any of the tabular data nor with Table 2.
Thank you for your suggestions. However, there is a limit of 200 words for the abstract, and the authors chose to present which findings were significantly impacted as opposed to their figures. If possible, we prefer to keep the abstract structure as-is. In addition, table 2 presents the proximate composition of the whole-body, not the BSF itself (Table 1). Modifications were made to improve clarity.
L.34- Add ‘with’ to provided with BSF or BSF: CF. Or replace provided with fed
Thank you for your correction. The text was modified accordingly.
L.38- Only insect larvae is accepted as keyword. Check the abstract very well and bring out the
Keywords
Thank you for your suggestion. We added Aquafeed, neotropical fish
L51- What is the protein content of BSF used in the project
The nutritional value of BSF is provided in table 1.
- 93 and L94 Not clear. Recast. What was your sample population and how many tanks were used? Indicate replication if any.
Thank you for your correction. The text was modified accordingly.
L.102- Add for (to account for the---
Thank you for your correction. The text was modified accordingly.
L.104- - L107 Not clear. Recast.
Thank you for your correction. The text was modified accordingly.
L.108 wrong spelling of tambaqui
Thank you for your correction. The text was modified accordingly.
L.112 Table 1.I do not agree with the centesimal computation and energy values of BSF and CF.
The raw data should be crosschecked especially with respect to crude protein value.
The authors don’t understand this comment. As stated in the material and methods section, the protein composition of the BSF was estimated by multiplying the total nitrogen by 5.25 instead of 6.25. This was to account for the chitin-nitrogen levels that are usually found in the BSF samples.
L.127-L.133-Why did none of the measured parameter values reflect on the abstract?.
Once again, the authors highlighted the study's findings due to the abstract's limited word count.
- L. 203- Table 2. Most computatonal values are wrong and should be recalculate, for instance,
Taken that initial wt is 115.2g and the final wts were 285.9g, 337.96g and 364.67g respectively
for BSF, BSF:CF and CF respectively. The daily weight gains would have been 1.4225g, 1.856g and
2.0789g rather than 140g, 11.83g and 2.40g indicated on the table.
So, there is need to recalculate all the tabular values for the other parameters
Thank you for your comment. The average initial weight of all tanks was 115.2 grams; therefore, it is expected that a difference of 0.02, 0.02, and 0.03 may occur per dietary treatment.
Under variables change Body weight 36 days (g), 68 days (g) and 98 days (g) to Body weight (g)
36 days, Body weight (g) 68 days and Bosy weight (g) 98 days, respectively
Thank you for your correction. The text was modified accordingly.
- 225 To the author’s knowledge change to authors’…
Thank you for your correction. The text was modified accordingly.
L-280- Interestingly, add a coma
Thank you for your correction. The text was modified accordingly.
L.295 Delete one fed
The authors don’t understand what the reviewer is asking.
L.308- Add d to provide
Thank you for your correction. The text was modified accordingly.
- 337, L.372, L.374 and L.375 add colon after In. In; R.W. Hardy, In: Division, In: nutrient … etc
Thank you for your correction. The text was modified accordingly.
- 424.- Colossoma Macropomum, ‘m’ should be in lower case.
Thank you for your suggestion. The authors are citing the title of the paper how it was published, even though the scientific name should not be capitalized. We hope the reviewer understands that this was not a mistake on our end.
I suggest all the scientific and journal names in the reference section be italicized except if not required by the journal.
Thank you for your correction. The text was modified accordingly.

Round 2
Reviewer 1 Report
Dear authors,
thanks to revised the manuscript. I' ve read the manuscript several times, but I am not sure that the experimental design is appropriated. Did the administrated "diets" fit the fish nutrional requirements in terms of amino acid and energy? Why authors did not mantain at least the metab. energy or protein level at the same level? How authors explain and discuss (not present now) the low FCR of BSF group? How can we compare a balanced diet (CF) with a single ingredient (whole BSF?)? What are the scientific basis to do this? I am really sorry because the idea is interesting, but it is almost impossible to compare the obtained data.